# MSA-VT Score for Assessment of Long-Term Prognosis after Electrical Storm Ablation

**DOI:** 10.3390/biomedicines12030493

**Published:** 2024-02-22

**Authors:** Radu Vatasescu, Cosmin Cojocaru, Viviana Gondos, Corneliu Iorgulescu, Stefan Bogdan, Sebastian Onciul, Antonio Berruezo

**Affiliations:** 1Department of Cardiothoracic Pathology, Faculty of Medicine, “Carol Davila” University of Medicine and Pharmacy, 050474 Bucharest, Romania; radu_vatasescu@yahoo.com (R.V.); stefan_n_bogdan@yahoo.com (S.B.); sebastian.onciul@gmail.com (S.O.); 2Cardiology Department, Clinical Emergency Hospital of Bucharest, 014461 Bucharest, Romania; iorgulescu_corneliu@yahoo.com; 3Department of Medical Electronics and Informatics, Polytechnic University, 060042 Bucharest, Romania; viviana.gondos@hellimed.ro; 4Cardiology Department, Elias University Emergency Hospital, 011461 Bucharest, Romania; 5Heart Institute, Teknon Medical Center, C/Vilana, 12, 08022 Barcelona, Spain; antonio.berruezo@quironsalud.es

**Keywords:** electrical storm, catheter ablation, risk assessment, mortality, recurrence

## Abstract

Introduction: Prognosis after electrical storm (ES) ablation remains severe, especially in patients with recurrent sustained monomorphic ventricular tachycardia (SMVT) or progressive heart failure (HF). However, single-factor-based prediction is suboptimal and may be refined by more complex algorithms. We sought to evaluate if a novel score MSA-VT (M = moderate/severe mitral regurgitation, S = severe HF at admission, A = atrial fibrillation at admission, VT = inducible SMVT after ablation) may improve prediction of death and recurrences compared to single factors and previous scores (PAINESD, RIVA and I-VT). Methods: A total of 101 consecutive ES ablation patients were retrospectively analyzed over a 32.8-month (IQR 10-68) interval. The MSA-VT score was calculated as the sum of the previously mentioned factors’ coefficients based on hazard ratio values in Cox regression analysis. The AUC for death prediction by MSA-VT was 0.84 (*p* < 0.001), superior to PAINESD (AUC 0.63, *p* = 0.03), RIVA (AUC 0.69, *p* = 0.02) and I-VT (0.56, *p* = 0.3). MSA-VT ≥ 3 was associated with significantly higher mortality during follow-up (52.7%, *p* < 0.001). Conclusions: Prediction by single factors and previously published scores after ES ablation may be improved by the novel MSA-VT score; however, this requires further external validation in larger samples.

## 1. Introduction

Electrical storm (ES) is defined as multiple (≥3), distinct (separated by more than 5 min) episodes of ventricular tachycardia (VT) or ventricular fibrillation (VF) clustered in a short interval of 24 h [1]. ES has emerged as a distinct clinical entity due to its significantly more severe clinical outcomes (in terms of death and heart failure (HF) hospitalizations) compared to those induced by isolated, infrequent episodes of VTs requiring adequate internal cardioverter defibrillator (ICD) interventions [1,2,3,4]. ES induces a three-fold higher risk of mortality which exceeds 30% during the first year after its occurrence, predominantly driven by HF [5,6]. Contemporary ES management is complex and multimodal: ICD programming optimization (to avoid unnecessary shocks), correction of precipitating triggers (drug side effects, electrolyte or metabolic disorders, ongoing myocardial ischemia), antiarrhythmic therapy (AAD), sedation, catheter ablation or autonomic modulation leading to short-term mechanical circulatory support [7].

Even if robust evidence supports the role of radiofrequency catheter ablation to improve outcomes in drug-refractory ES by [8,9,10,11,12], prognosis in certain subsets of patients may remain poor despite percutaneous treatment. Hence, it is critical to identify factors that aggravate outcomes to allow implementation of advanced HF therapies such as candidacy for heart transplant (HTx) or left ventricular assist devices (LVAD).

In this sense, there are certain factors (particularly VT recurrences, advanced HF, ES itself) [13,14] proven to affect prognosis after ablation for isolated episodes of VT. Moreover, even after VT ablation, experience regarding the role of scoring algorithms to refine event prediction is rather limited [15,16,17,18]. To date, few algorithms have directly sought to assess long-term mortality and recurrences after VT ablation (I-VT and MORTALITIES-VA) [16,18], whereas others have been developed to identify the need for periprocedural mechanical cardiocirculatory support (MCS) for acute heart failure (AHF) decompensation (PAINESD) [15] or the risk of complications and post-ablation 30-day mortality (RIVA) [17].

This is why this study seeks to develop and evaluate the role of a novel score to particularly assess the risk of death and/or recurrences in ES, where no previous algorithms have been specifically implemented.

## 2. Materials and Methods

### 2.1. Study Population

This retrospective study is based on a monocentric longitudinal analysis of baseline and post-procedural characteristics of consecutive ES patients, as described in previously published papers [19,20].
-A minimum of three separate episodes of sustained ventricular monomorphic tachycardia (SMVT) treated by adequate ICD therapies in a 24 h interval refractory to medical antiarrhythmic (AAD) treatment and without reversible triggers [21,22,23].-Treated by radiofrequency catheter ablation (RFCA) targeting ventricular arrhythmic substrate from January 2014 to December 2022.

### 2.2. Imaging, Electrophysiology Study and Ablation Strategy

The patients were considered elective if the ES episode occurred more than 7 days prior to current admission and was suppressed by AADs. ES episodes manifesting during the last 7 days before admission, which were pharmacologically stabilized and had no recurring VTs 24 h before ablation, were defined as acute stabilized ES. If VT episodes recurred during the 24 h prior to admission despite AADs, the case was considered to be acute ES. Ischemic cardiomyopathy (ICM) was considered if patients had a previous myocardial infarction. Patients with recurrent VTs caused by acute coronary syndromes were excluded from this cohort. Only characteristics from each patient’s most recent ablation procedure were reported.

Electrophysiological study (EPS) and ablation were performed in the fasting state and under conscious sedation and analgesia. A Boston Scientific Labsystem PRO EP Recording System v.2.7.0.16 was used for electrogram recording and analysis. High-density electroanatomical mapping with CARTO-3™ (Biosense Webster, Diamond Bar, CA, USA) (>1800 points, 70% of points emphasizing the scar area and its border zone (BZ)) was performed in sinus rhythm (SR) with 16–500 Hz signal filtering. Right ventricle catheterization was obtained by transfemoral access, whereas the left ventricle was instrumented by a trans-septal or retrograde aortic approach. The pericardial space was accessed by a fluoroscopy-guided anterior subxiphoid puncture. Remote magnetic navigation (RMN) (Niobe II, Stereotaxis Inc., St. Louis, MO, USA) and/or multielectrode catheter mapping (decapolar or duodecapolar) was used at the electrophysiologist’s case-specific decision. Mitral regurgitation (MR) severity and biplane Simpson-based left ventricular ejection fraction (LVEF) in transthoracic echocardiography was based on previous recommendations for diagnosis [24,25].

In endocardial mapping, normal myocardium was defined by local electrograms’ amplitude (EGMs) >1.5 mV (bipolar), >8.3 mV (unipolar LV), >5.5 mV (unipolar RV), whereas dense scar and border zone (BZ) myocardium were defined by endocardial bipolar signals’ amplitude < 0.5 mV and 0.5–1.5 mV, respectively. In epicardial mapping, bipolar EGM amplitude > 1 mV defined normal myocardium. Similarly to previous methodology, a scar-dechannelling protocol targeting conduction channel entrances (CCEs) [26] and using open-irrigated ablation catheters (35–50 W, 45 °C) was used. If VTs were spontaneously or mechanically induced and hemodynamically tolerated, activation/entrainment was performed. After scar-dechanelling, programmed electrical stimulation (PES) was routinely performed with at least 2 drive cycle lengths (CLs) and 4 extra stimuli (ESx) (3 ESx in patients with severe HF symptoms at rest or extreme frailty) (at a minimum of 200 ms or until ventricular refractoriness) from two sites (medially and laterally to the scar) to test residual inducibility (as previously described [19]). If SMVTs (which were considered relevant if their cycle lengths (CLs) ≥ 250 ms) [27] were induced, scar reconnection was reassessed and the scar-dechannelling protocol was repeated. Any SMVT induced during PES with CL ≥ 250 was considered to be residual inducibility at PES. Clinical SMVT was interpreted by 12-lead electrocardiogram QRS morphology or by ICD-derived intracardiac electrograms with similar (±20 ms) CLs.

### 2.3. Follow-Up Protocol

All follow-up data were reported in relation to the most recent ablation (in patients with multiple procedures) and were obtained from reviewing medical records and routine 6-month-interval ICD interrogations. For patients not evaluated in our center, data were obtained from referring physicians regarding the clinical course and ICD interrogations. ICD interrogation was performed in all patients that were alive in June 2023.

All-cause mortality rates were retrospectively analyzed after the moment of the most recent ablation, irrespective of cardiovascular and non-cardiovascular causes of death. Recurrences were defined by SMVT or VF episodes adequately treated by ICD intervention (either antitachycardia pacing or shock). ICD detection intervals were programmed to allow detection of any ventricular arrhythmia which was previously spontaneously present induced at PES (−20 bpm relative to the slowest known VT), and no monitoring zones were set to assess for arrhythmias with longer CLs.

The study protocol adhered to the principles of the Declaration of Helsinki and was approved by the human research committee of the Emergency Clinical Hospital of Bucharest Ethics Committee (12521–1 April 2022).

### 2.4. Endpoints

This study sought to evaluate if a new scoring algorithm (MSA-VT—moderate/severe functional mitral regurgitation, Severe NYHA III/IV heart failure, AF at admission and residually inducible sustained monomorphic VT at end-procedural PES) improves prediction of all-cause mortality and VT/VF recurrence after ES ablation in our cohort, compared to previously existing scores (PAINESD [15,28], I-VT [16] and RIVA scores [17,29]).

Based on the number of events observed during follow-up (deaths and recurrences), four factors were selected for development of the score. All parameters which had significantly different distributions (for non-numerical variables) or mean values (for numerical variables) in alive versus dead subgroups of the cohort (Appendix A) were selected for Cox regression analysis. Factors which provided statistically significant prediction (*p* < 0.05) in univariate Cox regression were selected for development of this score. A coefficient was attributed to each factor based on its HR (in univariate Cox regression analysis) by rounding to the nearest integer; the score was based as the sum of the factors’ coefficients (5 points were attributed for residually inducible SMVT, 3 points for NYHA III-IV at admission, 4 points for moderate/severe MR, 4 points for AF at admission).

### 2.5. Statistical Analysis

Continuous data were expressed as mean ± standard deviation (SD) for normally distributed data and median (IQR) for non-normally distributed data. Categorical data were expressed as percentages (counts). The normality of data was evaluated by Kolmogorov–Smirnov tests. Categorical variables were compared using Fisher’s exact test/chi-square analysis, and continuous variables were compared using Student’s *t*-test (if normally distributed) and non-parametric tests (Mann–Whitney U test).

ROC curve analysis was performed to compare and graphically display area under curve (AUC) values for prediction of death and recurrences during follow-up of the MSA-VT novel score versus single risk factors and versus PAINESD/I-VT/RIVA scores, respectively. Optimal cut-off value on ROC curve analysis was selected based on Youden’s index. Internal validation of the scoring algorithm was performed on a subpopulation randomly selected by the statistical software representing 75% of the entire analyzed cohort. Survival curves during follow-up were plotted by the Kaplan–Meier method and pairwise over-strata comparison was determined using log–rank test.

A 2-sided *p*-value < 0.05 was considered statistically significant. Statistical analysis was performed using SPSS version 23 (IBM Corp., Armonk, NY, USA) software and Prism 9 Version 9.5.0 (GraphPad Software, LLC, San Diego, CA, USA).

## 3. Results

One hundred and one (n = 101) consecutive patients were included and monitored for 32.8 months (IQR 25–75% 10–68). Table 1 summarizes patient characteristics. In the non-ischemic cardiomyopathy subgroup (NICM), there were nine (27.2%) patients diagnosed with arrhythmogenic cardiomyopathy, four (12.1%) patients with non-compaction cardiomyopathy, four (12.1%) patients with valvular cardiomyopathies, six (5.9%) post-myocarditis cardiomyopathy, ten (9.9%) patients with idiopathic DCM. Twenty-four (23.8%) patients had a history of paroxysmal AF, eight (7.9%) of persistent AF and eight (7.9%) of permanent AF. Patients from the ischemic cardiomyopathy (ICM) subgroup were older (*p* = 0.002) and were more frequently hypertensive (*p* = 0.002), dyslipidemic (*p* < 0.001), smokers (*p* = 0.02) or with chronic kidney disease (*p* = 0.005). ICM patients had a higher PAINESD score (*p* = 0.001) but similar I-VT (*p* = 0.56) and RIVA (*p* = 0.3) scores. There were 41.6% (n = 42) patients in PAINESD class III, 37.6% (n = 38) in PAINESD class II and 20.8% (n = 21) patients in PAINESD class I.

Table 2 summarizes the procedural characteristics and post-ablation follow-up data. In total, there were n = 139 ablation procedures performed for the n = 101 included patients. There were 31.7% (n = 32) patients with previous ablations (one with three procedures, five with two procedures and twenty-six with one previous procedure). There were 30.7% (n = 31) deaths and 35.6% (n = 36) patients with SMVT/VF recurrences during follow-up (32.8 months (10.06–68.91)). The median time-to-death was 24 months (9.5–42) and the median time-to-recurrence was 3.25 months (0.5–17.25).

Appendix A (in the Appendix A section of the manuscript) summarize the relation between categorical and numerical parameters, respectively, and the rate of all-cause mortality during follow-up. The rates of death during follow-up were significantly higher in those with AF at admission (69.2%) or a history of AF (47.5), with residually inducible SMVT at PES (59.4%), with moderate/severe MR (54.5%) and with NYHA III-IV at admission (51.7%). Patients that died during follow-up were older (65.5 vs. 57 years, *p* = 0.002) and with lower LVEF (28.7% vs. 33.4%, *p* = 0.03).

Table 3 summarizes the results of univariable Cox regression analysis regarding all-cause mortality predictors after ES ablation. Additionally, Cox regression multivariable model analysis including the parameters of the MSA-VT score demonstrated that residually inducible SMVT at PES (HR 3.9, CI 95% 1.8–8.4), NYHA III-IV at admission (HR 4.7, CI 95% 2–10.6), and moderate/severe MR (HR 5.5, CI 95% 2.4–12.2) were independent predictors of death during follow-up, whereas AF at admission (HR 1.7, CI 95% 0.7–4.1) did not independently predict death during follow-up.

Figure 1 shows ROC curve analysis for overall mortality during follow-up based on MSA-VT score (AUC 0.84, *p* < 0.001) compared to PAINESD score (AUC 0.63, *p* = 0.03), I-VT score (0.56, 0.3) and RIVA score (AUC 0.69, *p* = 0.02), respectively. ROC curve analysis for 30-day post-discharge mortality during follow-up showed the following AUC values: MSA-VT score (AUC 0.863, *p* < 0.014), MR (AUC 0.701, *p* = 0.17), AF at admission (AUC 0.562, *p* = 0.67), NYHA at admission (AUC 0.679, *p* = 0.22), age (AUC 0.658, *p* = 0.28), LVEF (AUC 0.458, *p* = 0.77), PAINESD (AUC 0.709, *p* = 0.001), RIVA (AUC 0.753, *p* = 0.08) and I-VT score (AUC 0.749 *p* = 0.09). Internal validation of ROC analysis parameters on a 75% randomized sample showed the following AUC values for predicting death during follow-up: PAINESD (0.613, *p* = 0.1), RIVA score (0.7, *p* = 0.003), I-VT score (0.522, *p* = 0.7) and MSA-VT score (0.83, *p* < 0.001). Appendix A shows ROC curve analysis used for comparison of death prediction during follow-up by MSA-VT score compared to singular risk factors (age, mitral regurgitation, LVEF, NYHA class, VT/VF recurrence). 

Figure 2 displays Kaplan–Meier survival curves stratified by an MSA-VT score of 3 points. Patients with an MSA-VT ≥ 3 points had a significantly higher mortality during follow-up compared to those with MSA-VT < 3 points (52.7% (n = 29) vs. 4.3% (n = 2), *p* < 0.001). Furthermore, recurrences during follow-up were also significantly more frequent in patients with MSA-VT ≥ 3 points (49.1% (n = 27) vs 19.6% (n = 9), *p* = 0.003). Appendix A provides a detailed Kaplan–Meier analysis for survival during follow-up stratified by RIVA, PAINESD and I-VT classes based on each scoring algorithm’s reported optimal cut-off value.

Univariable Cox regression demonstrated that residually inducible SMVT predicted recurrences (HR 6.5, *p* < 0.001) and that NYHA III-IV predicted recurrences (HR 2.4, *p* = 0.009). Moderate/severe MR and AF at admission did not predict recurrences (*p* = 0.15 and *p* = 0.12, respectively). ROC curve analysis for recurrence during follow-up showed the following AUC values: MSA-VT score (AUC 0.72, *p* < 0.001), MR (AUC 0.58, *p* = 0.17), AF at admission (AUC 0.550, *p* = 0.413), NYHA at admission (AUC 0.627, *p* = 0.037), age (AUC 0.578, *p* = 0.198), LVEF (AUC 0.405, *p* = 0.115), PAINESD (AUC 0.580, *p* = 0.184), RIVA (AUC 0.570, *p* = 0.24) and I-VT (AUC 0.599, *p* = 0.09).

There were 11.8% complications (n = 12): one brachial artery thromboembolism, six cases of pericardial effusions (none requiring pericardiocentesis), one transient episode of coronary ischemia by spasm, one thromboembolic stroke and three vascular access hematomas. There was no intraprocedural mortality. However, there were two deaths during hospitalization (one developed refractory HF five days after ablation and one was due to sepsis).

## 4. Discussion

### 4.1. MSA-VT to Predict Death and Recurrences in Comparison with Previous Scoring Algorithms

Numerous factors have proven reliable in assessing periprocedural risk and anticipating prognosis after VT ablation (however not exclusively in ES) [4,8,10,11,13,14,19,30,31].

The PAINESD scoring algorithm was initially developed and validated to identify patients at risk for hemodynamic deterioration which may benefit from periprocedural mechanical cardio-circulatory support (MCS) [15,28]. According to PAINESD, elderly diabetic patients with ICM and low LVEF that develop ES and HF have the highest risk of developing cardiogenic shock. Subsequent studies have indicated its ability to also predict mortality VT ablation, yet certain studies have reported conflicting results regarding its accuracy, particularly regarding recurrences [32,33,34]. Existing data demonstrate different clinical trajectories in Kaplan–Meier or Cox regression in different PAINESD classes, yet no ROC curve analyses are available. Even if 42.6% of patients were classified as high risk by PAINESD scores in our dataset, there were no cases of intraprocedural acute heart failure (AHF). This is most likely explained by sinus rhythm-based mapping and ablation strategies that decrease intraprocedural VT burden and the associated risks of acute heart failure (AHF), especially in the context of ES [35], which differs from protocols applied in the PAINESD study, which relied on VT induction and general anesthesia (used in up to one-third of MCS patients) [15]. Additionally, only 5% of ES patients in our dataset required acute ablation (<24 h), which is known to alter prognosis [30]. Furthermore, as detailed in the Appendix A section, the only statistically different PAINESD classifications in terms of long-term mortality were low risk (<8 points) versus high risk (>15 points); this suggests that, at least in this particular ES cohort, PAINESD may not adequately stratify clusters of patients with different mortality trajectories.

The RIVA score was designed to assess the risk of procedural complications and in-hospital mortality and received changes in the modified RIVA variant (mRIVA) [17]. In this regard, it attributes incremental severity to patients with specific structural heart disease (particularly ICM), the need for more complex procedures (i.e., epicardial), especially in patients receiving antithrombotic treatment and suffering from renal dysfunction [17]. To our knowledge, this paper provides the first evaluation of the RIVA score on long-term outcomes. Due to the significantly lower number included in this cohort (compared to 1417 subjects enrolled in the RIVA study [17]), this analysis only dichotomized patients in the low-risk category (<13 points) and non-low-risk category (≥13 points) which demonstrated significantly different patterns of mortality during follow up (Appendix A). Notably, Mathew et al. [17] reported a total of 11 different clusters with incremental risks of complications and in-hospital mortality. In our study, the RIVA score demonstrated better prediction of overall death during follow-up compared to PAINESD and the I-VT score, but was, however, not accurate in predicting recurrences. Additionally, MSA-VT was the only scoring test to adequately predict recurrences in our dataset (although it was designed to predict mortality).

The scoring algorithms quantify risk at different moments in relation to the ablation procedure. PAINESD, RIVA and I-VT scores aid in preprocedural risk assessment, whereas MSA-VT is conditioned by post-ablation results (as is the post-procedural variant of I-VT). This version of I-VT score calculates mortality risk based on LVEF above or below 30%, inducibility at PES, presence of diabetes mellitus, age higher than 80 years and the presence of ES. However, this algorithm did not adequately predict death during follow-up in this particular cohort.

In our dataset, MSA-VT score appears to provide a better prediction of events after ES ablation in terms of both mortality (during long-term follow-up and during the first 30-days after discharge) and recurrences in comparison to the aforementioned algorithms. Clinically applied, this analysis suggests two different clinical profiles of ES patients treated by ablation. Any patient affected by least one of these factors (the inability to eliminate all VT morphologies by ablation, HF decompensation at admission, AF and/or moderate or severe MR) will score higher than 3 points in the MSA-VT algorithm and should be considered at high risk for subsequent events. Conversely, in ES patients with less advanced structural heart disease (not affected by overt HF, maintaining sinus rhythm and with no significant MR) in which ablation eliminates all arrhythmogenic myocardial substrate and leads to non-inducibility at PES, long-term survival without recurrent events can be expected.

In conclusion, the MSA-VT score may provide a novel post-procedural tool to evaluate the long-term risk of death and recurrences in ES patients. However, it requires external validation in larger samples to further assess its prognostic capacity.

### 4.2. MSA-VT Risk Factors’ Effect on Long-Term Outcomes

The MSA-VT score only partially shares prediction parameters with previously published algorithms: residually inducible SMVT after ablation (shared only with the post-ablation variant of I-VT) and NYHA III-IV at admission (shared with PAINESD and mRVA), which have been thoroughly evaluated in multiple previous papers and are known prognostic factors [10,11,19,36]. Notably, in our cohort, mortality was highest in patients with either residual SMVT and severe HF at admission (59.4% and 51.4% respectively) which is in line with outcomes reported in larger samples [10,11,36]. Translating into clinical practice, we consider that lack of control of either HF or post-ablation VTs in ES patients should mandate swift implementation of MCS or referral to heart transplant.

The presence of AF at admission and moderate/severe MR were not previously assessed as prognostic markers after VT ablation. Meta-analysis data demonstrates that ICD patients with AF have an approximately two-fold higher risk of death and appropriate ICD therapies compared to those in sinus rhythm [37]. Furthermore, AF itself may induce ventricular arrhythmias in ICD recipients [38]. MR also impacts VT burden in addition to its role in HF development and pump failure progression [39]. Recent studies have shown reduction in ICD therapies in patients successfully treated by percutaneous mitral valve repair (PMVR) [40,41,42,43]. In our dataset, patients in AF at admission displayed very high mortality (69.2%), similarly to those with moderate/severe MR (54.5%). Hence, we considered them plausible predictors of long-term clinical course. Moderate/severe MR and AF at admission provided 4.2-fold higher and 3.5-fold higher, respectively, risk of death during follow-up in univariable Cox regression. In a four-variable model, only residually inducible SMVT, NYHA III-IV and moderate/severe MR independently predicted death and not AF at admission. Additionally, MR and AF at admission were not able to predict recurrences in our dataset.

## 5. Limitations


This analysis is based on a relatively small-scale sample of 101 patients. Previous studies that assessed the role of scoring algorithms analyzed samples ranging from approximately 175–193 patients (MORTALITIES-VA and PAINESD scores) up to 1251–1417 patients (I-VT and RIVA scores). However, all of these studies included VT patients treated by ablation (and not exclusively ES).The findings of this study are based on a single-center population analysis. External validation of the newly proposed scoring algorithm (MSA-VT) on larger samples from distinct ablation centers is required. Internal validation was based on selection of a randomized sub-sample of 75% of the initial cohort and showed higher values of AUC compared to previously mentioned scores.Post-ablation PES was based on a four-stimulus protocol in only 66.3% of patients, which may influence residual VT inducibility. However, this is in line with large-scale studies regarding uniformity of the PES protocol [10,11].No mechanical circulatory support (MCS) was available during this study. This is a significant difference in protocol compared to previously published experience, (in particular, the PAINESD study). However, most patients were previously stabilized (thus optimized in terms of HF) and subsequently ablated (predominantly by substrate ablation during sinus rhythm).


## 6. Conclusions

In our cohort, the MSA-VT score improved the prediction of survival and recurrence in comparison to available validated risk assessment scores.

## Figures and Tables

**Figure 1 biomedicines-12-00493-f001:**
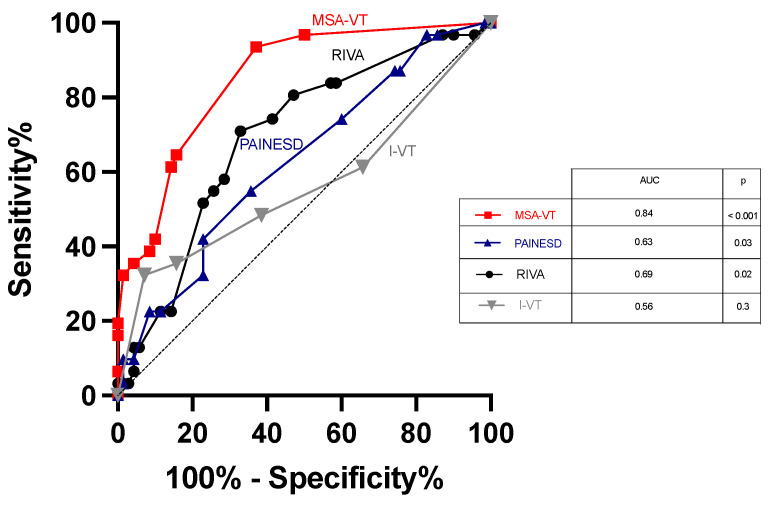
Receiver–operator curve analysis for prediction of death during follow-up based on MSA-VT score compared to RIVA, PAINESD and I-VT scores.

**Figure 2 biomedicines-12-00493-f002:**
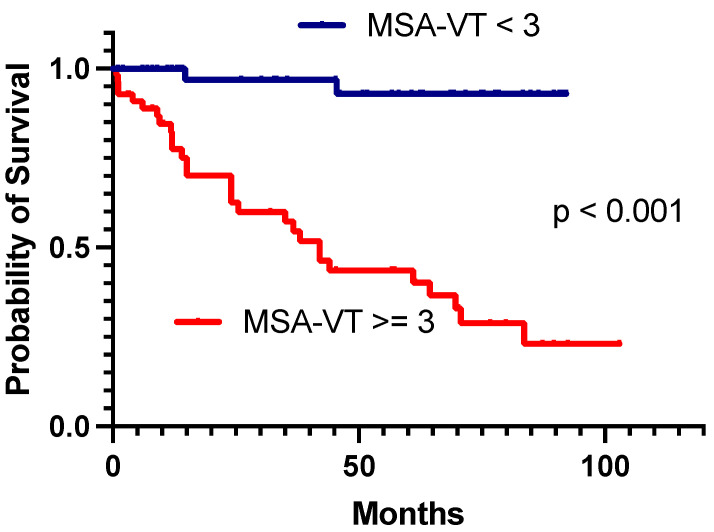
Kaplan–Meier survival curves during follow-up dichotomized by MSA-VT score higher than or equal to three points versus lower than three points, showing significantly different rates of mortality during post-ablation follow-up.

**Table 1 biomedicines-12-00493-t001:** Baseline patient characteristics prior to the moment of ablation. SD = standard deviation, BB = beta blocker, ES = electrical storm, ICD = implantable cardioverted defibrillator, AF = atrial fibrillation, LVEF = left venrticular ejection fraction, MR = mitral regurgitation, NYHA = New York Heart Association, CRT = cardiac resynchronization therapy, COPD = chronic obstructive pulmonary disease, BMI = body mass-index, T2DM = type 2 diabetes mellitus. Bold font formatting highlights the variables with significant difference (*p* < 0.05) between the ischemic cardiomyopathy and the non-ischemic cardiomyopathy subgroups.

	Overall	Ischemic Cardiomyopathy (67.3% (n = 68))	Non-Ischemic Cardiomyopathy (32.7% (n = 33))	*p*
Males, % (n)	86.1% (n = 87)	86.8% (n = 59)	84.8% (n = 28)	0.76
**Age, mean (SD)**	**59.6 ± 12.8**	**62.4 ± 11.1**	**54 ± 14.4**	**0.002**
BMI ≥ 25 kg/sqm, % (n)	34.7% (n = 35)	35.3% (n = 24)	33.3 (n = 11)	0.99
COPD, % (n)	8.9% (n = 9)	7.4% (n = 5)	12.1% (n = 4)	0.46
T2DM, % (n)	62.4% (n = 63)	30.9% (n = 21)	15.2% (n = 5)	0.14
**Hypertension, % (n)**	**62.4% (n = 63)**	**73.5% (n = 50)**	**39.4% (n = 13)**	**0.002**
**Dyslipidemia, % (n)**	**61.4% (n = 62)**	**75% (n = 51)**	**33.3% (n = 11)**	**<0.001**
**Smoker, % (n)**	**18.8% (n = 19)**	**25% (n = 17)**	**6.1% (n = 2)**	**0.02**
**CKD, % (n)**	**18.8% (n = 19)**	**26.5% (n = 18)**	**3% (n = 1)**	**0.005**
Previous CRT, % (n)	14.9% (n = 15)	11.8% (n = 8)	21.2% (n = 7)	0.24
BB prior to ES	82% (n = 83)	80.% (n = 55)	84.8% (n = 28)	0.78
BB after ES ablation	86.1% (n = 87)	85.3% (n = 58)	87.9% (n = 29)	0.99
Amiodarone prior to ES	68.3% (n = 69)	61.8% (n = 42)	78.8% (n = 26)	0.11
Amiodarone after ES ablation	71.3% (n = 72)	66.2% (n = 45)	81.8% (n = 27)	0.15
Weeks from ES to ablation	1.76 ± 2.85	1.8 ± 2.6	1.6 ± 3.2	0.85
Number of ICD therapies, mean (SD)	14.12 ± 23.66	12.7 ± 16.5	16.5 ± 32.9	0.59
Admission creatinine, mean (SD)	1.11 ± 0.59	1.16 ± 0.7	1.02 ± 0.2	0.27
AF at admission, % (n)	12.9% (n = 13)	13.2% (n = 9)	12.1% (n = 4)	0.99
History of AF, % (n)	39.6% (n = 40)	35.3% (n = 24)	48.5% (n = 16)	0.27
LVEF, mean (SD)	32% ± 11.6	31.5 ± 10.8	33 ± 13.4	0.55
Moderate/severe MR, % (n)	32.7% (n = 33)	35.3% (n = 24)	27.3% (n = 9)	0.5
NYHA III-IV at admission, % (n)	28.7% (n= 29)	26.5% (n = 18)	33.3% (n = 11)	0.49
**PAINESD score, mean (SD)**	**14.5 ± 6.11**	**16.8 ± 5**	**9.9 ± 5.5**	**<0.001**
Postprocedural I-VT risk score for death, mean (SD)	1.1 ± 1.6	1.04 ± 1.49	1.24 ± 1.83	0.56
RIVA score, mean (SD)	12.3 ± 4.7	12.6 ± 4.6	11.6 ± 5	0.3
MSA-VT score, mean (SD)	4.26 ± 4.24	4.13 ± 3.96	4.54 ± 4.82	0.64

**Table 2 biomedicines-12-00493-t002:** Procedural and post-ablation follow-up characteristics. SMVT = sustained monomorphic ventricular tachycardia, PES = programmed electrical stimulation, IQR = interquartile range, VF = ventricular fibrillation, ESx = extrastimuli, SD = standard deviation. Bold font formatting highlights the variables with significant difference (*p* < 0.05) between the ischemic cardiomyopathy and the non-ischemic cardiomyopathy subgroups.

	Overall	Ischemic Cardiomyopathy (67.3% (n = 68))	Non-Ischemic Cardiomyopathy (32.7% (n = 33))	*p*
Previous ablation procedures, % (n)	31.7% (n = 32)	27.9% (n = 19)	39.4% (n = 13)	0.26
Substrate mapping & ablation, % (n)	94.9% (n = 93)	97% (n = 65)	90.3% (n = 28)	0.32
**Activation mapping, % (n)**	61.4% (n = 62)	54.4% (n = 37)	75.8% (n = 25)	**0.05**
Number of SMVTs induced during the procedure mean (SD)	2.26 ± 0.1	2.16 ± 2.1	2.45 ± 1.65	0.52
Remote magnetic navigation mapping & ablation, % (n)	78.2% (n = 79)	77.9% (n = 53)	78.8% (n = 26)	0.99
**Endocardial + epicardial ablation, % (n)**	20.8% (n = 21)	8.8% (n = 6)	45.5% (n = 15)	**<0.001**
Ablation type—acute, % (n)	5% (n = 5)	5.9% (n = 4)	3% (n = 1)	0.69
Ablation type—stabilized, % (n)	80.2% (n = 81)	77.9% (n = 53)	84.8% (n = 28)	
Ablation type—elective, % (n)	14.9% (n = 15)	16.2% (n = 11)	12.1% (n = 4)	
4-ESx PES, % (n)	66.3% (n = 67)	69.7% (n = 47)	60.6% (n = 20)	0.37
Endocardial mapping points (median, IQR)	1900 (1100–2455)	1983 (1310.5–2546.5)	1750 (589.75–2165.2)	0.11
**Ablation points, (IQR)**	37 (22–57)	47.5 (28–61)	27 (16–42)	**0.001**
Multielectrode catheter mapping, % (n)	27% (n = 26.7)	26.4% (n = 18)	27.3% (n = 9)	0.99
Residual SMVT inducible at PES after ablation, % (n)	31.7% (n = 32)	27.9% (n = 19)	39.4% (n = 13)	0.26
**Days of hospitalization, mean (SD)**	10 ± 9.9	8.1 ± 6.9	14 ± 13.6	**0.028**
All-cause mortality during follow-up, % (n)	30.7% (n = 31)	27.9% (n = 19)	36.4% (n = 12)	0.49
Post-discharge 30-days mortality, % (n)	4% (n = 4)	4.4% (n = 30	3% (n =1)	0.99
SMVT/VF recurrences during follow-up, % (n)	35.6% (n = 36)	29.4% (n = 20)	48.5% (n = 16)	0.077

**Table 3 biomedicines-12-00493-t003:** Cox regression univariable analysis regarding the risk of death during follow-up. HR = hazard ratio, PES = programmed electrical stimulation, NYHA = New York Heart Association, AF = atrial fibrillation, CKD = chronic kidney disease, T2DM = type 2 diabetes mellitus, COPD = chronic obstructive pulmonary disease, VT = ventricular tachycardia. Bold font formatting highlights the variables with significant difference (*p* < 0.05) between the ischemic cardiomyopathy and the non-ischemic cardiomyopathy subgroups.

Parameter	HR	CI 95%	*p*
**Residually inducible SMVT at PES after ablation**	**4.9**	**2.3–10.2**	**<0.001**
**NYHA III-IV at admission**	**3.2**	**1.5–6.2**	**0.002**
**Moderate/severe MR**	**4.2**	**2–8.7**	**<0.001**
**AF at admission**	**3.5**	**1.6–7.8**	**0.002**
Ischemic cardiomyopathy	1.07	0.5–2.2	0.85
**Age**	**1.06**	**1.02–1.1**	**0.001**
LVEF < 25%	1.7	0.8–3.6	0.11
COPD	2	0.7–6	0.16
T2DM	1.5	0.7–3.1	0.28
CKD	2.4	0.9–6.2	0.051
Total number of ablation procedures	1.1	0.6–2.1	0.6
Number of VTs induced during the procedure	1.1	0.9–1.2	0.054

## Data Availability

Study data are available upon request by any third parties from the corresponding author.

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
