# Peer review of "MSA-VT Score for Assessment of Long-Term Prognosis after Electrical Storm Ablation"

_biomedicines, 2024, doi:10.3390/biomedicines12030493_

Round 1

Reviewer 1 Report

Comments and Suggestions for Authors

The  study of  novel MSA-VT score validation  in  patients  who underwent  VT ablation was  presented.   The  study is  retrospective in its  nature, however sufficient  for  current  conclusions. Moreover, the  current  paper continues the  series  from  this  authors  group.

Abstract  represents  the  body of  the  paper.

Introduction section could  be  improved. Its recommended  to present  the  scope  of  the  problem of ventricular storm  (VS), including  definition, epidemiology prognosis and current  approaches. Radiofrequent  ablation  significantly change  prognosis in  many of  these patients,  however  mortality rate  is  still high.

Material and  methods.  Patients  characteristics  and  procedure  was well described with  links  to previous  papers.  Did the  patient  population include  VS  in patients  with  acute myocardial infarction? Combined  endocardial-epicardial  ablation were not utilized?

 In the  endpoint  subsection more  data about  MSA-VT score is  required such as  number of points  ets (move  from statistics section). Otherwise, some  data should  be  included into introduction section.

Statistics is  good.

Results.  Table  1  could  include  data on  MSA-VT score  eighter. Table  3  and  4  could  be  presented is  supplementary files.   

Was  multivariable  analysis attempted to be performed?

Figure  1 have  to be  mentioned in the  Result section.  Figure  1  A  could  be  moved  to the  supplementary  file.  Figure  1 B  is   the central illustration  of  the  study.

Figure  1n  C  have  to  be  mentioned in the  text.  Authors could dichotomize prognosis  based  on MSA-VT score  3, have  to be  discussed.

Discussion is  good. Limitations  section could  include  combined  epi-endocardial ablation.`

Comments on the Quality of English Language

Proofreading  is required

Author Response

Dear Reviewers,

We wish to thank you for your recommendations and hereby provide a point-by-point response to all of the observations we have received. All of these changes have also been implemented in the main manuscript (with tracked changes function) which we have uploaded in its new version. We have included all the comments and changes to the manuscript in the same letter in order to ease the process of revision. Black font marks the reviewers’ observation and our comments regarding the changes, whereas green font marks the specific changes we have applied to the text. We hope to have met the reviewers’ demands and hope to have improved the manuscript significantly.

Please find the Letter of Response to Reviewers hereby attached.

Reviewer 2 Report

Comments and Suggestions for Authors

1. The "Introduction" section should be significantly expanded. In the presented form, the necessary information is missing to understand the problem.

2. Tables 1, 2, 3 are not relevant to the essence of the work. They only overload the manuscript with essentially unnecessary information.

3. The authors should show in more detail the calculation of the proposed new scoring algorithm (MSA-VT). This is not clearly described in the current version of the article.

4. I recommend that the authors compare the algorithms (MSA-VT, RIVA, PAINESD, I-VT) with each other not only by AUC. To construct Kaplan Meier curves for all algorithms and compare them.

5. The Discussion section also requires a broader comparative analysis and interpretation of the results obtained in comparison with other existing data in this area.

Author Response

(The authors gave the same response as above.)

Reviewer 3 Report

Comments and Suggestions for Authors

Dear authors, I was reviewing with high interest the manuscript entitled "MSA-VT score for assessment of long-term prognosis after electrical storm ablation". You deal with a highly interesting subject and offer a real new score for risk prediction in these patient group. The literature is actual and recent and the discussion deals with all important aspects. However, one issue needs to be solved: The amount of interventions (e.g. sum of energy) per patient needs to be reported. It's a single center data collection and there is no way to compare the sum of applied interventions between different studies. In conclusion, this manuscript adds a new potent score for risk prediction in these patients.

Author Response

(The authors gave the same response as above.)

Round 2

Reviewer 1 Report

Comments and Suggestions for Authors

The paper was  improved substantially. Could  be  recommended  to the  Journal after  editing.

Author Response

Dear Reviewer 1,

We thank you for your observations and we are glad to have adequately implemented your insightful recommendations.

With great respect,

The Authors

Reviewer 2 Report

Comments and Suggestions for Authors

The authors fixed everything quickly and efficiently. Wonderful!

Author Response

Dear Reviewer 2,

We thank you for your observations and we are glad to have adequately implemented your insightful recommendations.

With great respect,

The Authors